# YOLOv11-XRBS: Enhanced Identification of Small and Low-Detail Explosives in X-Ray Backscatter Images

**DOI:** 10.3390/s25196130

**Published:** 2025-10-03

**Authors:** Baolu Yang, Zhe Yang, Xin Wang, Baozhong Mu, Jie Xu, Hong Li

**Affiliations:** 1MOE Key Laboratory of Advanced Micro-Structured Materials, School of Physics Science and Engineering, Tongji University, 1239 Siping Road, Shanghai 200092, China; yangbl@tongji.edu.cn (B.Y.); 2230984@tongji.edu.cn (Z.Y.); 1310581@tongji.edu.cn (J.X.); 2National Key Laboratory of Science and Technology on Near-surface Detection, Wuxi 214000, China; li_hong_09@126.com

**Keywords:** X-ray backscatter, YOLOv11-XRBS, explosive identification

## Abstract

Identifying concealed explosives in X-ray backscatter (XRBS) imagery remains a critical challenge, primarily due to low image contrasts, cluttered backgrounds, small object sizes, and limited structural details. To address these limitations, we propose YOLOv11-XRBS, an enhanced detection framework tailored to the characteristics of XRBS images. A dedicated dataset (SBCXray) comprising over 10,000 annotated images of simulated explosive scenarios under varied concealment conditions was constructed to support training and evaluation. The proposed framework introduces three targeted improvements: (1) adaptive architectural refinement to enhance multi-scale feature representation and suppress background interference, (2) a Size-Aware Focal Loss (SaFL) strategy to improve the detection of small and weak-feature objects, and (3) a recomposed loss function with scale-adaptive weighting to achieve more accurate bounding box localization. The experiments demonstrated that YOLOv11-XRBS achieves better performance compared to both existing YOLO variants and classical detection models such as Faster R-CNN, SSD512, RetinaNet, DETR, and VGGNet, achieving a mean average precision (*mAP*) of 94.8%. These results confirm the robustness and practicality of the proposed framework, highlighting its potential deployment in XRBS-based security inspection systems.

## 1. Introduction

The growing threat of improvised explosive devices has become a critical concern for public safety, border security, and military operations. The rapid and accurate detection of concealed explosives is essential in such contexts [1,2]. Although transmission X-ray systems are widely used, they primarily rely on material density and often struggle to detect low-atomic-number (low-Z) organic compounds, which are frequently present in explosive materials. XRBS imaging provides an effective alternative by capturing scattered radiation from the incident side [3]. This technique is particularly sensitive to low-Z elements such as carbon, hydrogen, oxygen, and nitrogen due to their relatively high Compton scattering cross-sections compared to high-Z materials. XRBS imaging offers key advantages in field deployment, including mobility, non-invasiveness, and real-time imaging, making it well suited to dynamic security screening tasks.

For XRBS technology, the development of intelligent detection methods specifically tailored to this modality remains limited [4]. Most existing XRBS systems have focused on mechanical and imaging enhancements, such as collimator design, dual-detector configurations, and beam modulation techniques [5,6]. In contrast, research on the recognition of XRBS images remains very limited. Compared to X-ray transmission imaging, which relies on material density to detect objects, XRBS provides better contrast for low-Z elements. However, X-ray transmission imaging struggles with detecting low-Z organic compounds, commonly found in explosives, especially small explosive devices, whose density often closely matches that of surrounding materials. Similarly, infrared imaging, while valuable in detecting temperature differences, faces challenges in identifying small, irregularly shaped explosive devices, particularly when they are partially occluded or embedded in cluttered backgrounds. For instance, Fourier-transform infrared (FTIR) spectroscopy has been applied for explosives detection but faces limitations such as background interference and the need for direct contact with the sample [7]. Most existing methods used to recognize explosives rely on deep learning [8], and such approaches were explored in X-ray transmission imaging [9], particularly with object detection architectures such as You Only Look Once (YOLO) and Faster Region-based Convolutional Neural Network (R-CNN) [10,11,12]. However, directly applying these models to XRBS data is challenging. XRBS images differ markedly in their contrast and spatial characteristics due to their Compton scattering effects and scanning geometry, which often leads to a reduction in the performance of pretrained models. Additionally, small explosive devices are frequently partially occluded or embedded within cluttered backgrounds, further diminishing their already limited visual features and making them particularly difficult to identify using conventional object detection methods.

To address these challenges, several issues must be addressed. XRBS images are often of low quality due to their weak signal strength and high background noise. Identifying small objects is critical, as explosive devices typically occupy a minimal portion of the image. Moreover, explosives often lack distinct visual features and may appear as irregular shapes, requiring identification algorithms to be robust to such variability [13]. Although state-of-the-art identify models—including YOLO variants, Transformer-based detectors, and Faster R-CNN—perform well on standard datasets, their effectiveness on XRBS images is limited [14,15]. This is largely due to the low contrast, small object sizes, and indistinct features characteristic of XRBS images, which differ significantly from the natural images used to train these models [16,17]. Additionally, the scarcity of publicly available, high-quality annotated XRBS datasets further hinders the development and benchmarking of specialized algorithms [18,19]. These limitations underscore the need for a detection framework specifically optimized for XRBS-based explosive identification. While the ultimate goal of our research is the identification of improvised explosive devices (IEDs), the present work employs inert replicas of factory-made explosive models (e.g., grenades) as representative and challenging targets. XRBS-based explosive detection fundamentally depends on recognizing material composition and shape contours; therefore, these prefabricated models provide an exploratory model for validating our methodology. Once feasibility is established, the framework can be extended in future studies to disguised or improvised explosive devices.

In this study, we address the gaps in the literature, focusing on the automated detection of explosive devices in XRBS images. First, we establish a dedicated dataset of 10,000 XRBS images, which provides both the foundation for our experiments and a valuable resource for subsequent research. Second, we propose an enhanced detection framework, YOLOv11-XRBS, which incorporates adaptive architecture refinement, the SaFL strategy, and a recomposed loss function to better handle the low-resolution, low-detail characteristics of XRBS imagery. Finally, we conduct comprehensive experiments to evaluate the model against multiple baselines and detection scenarios, demonstrating that YOLOv11-XRBS achieves higher reliability and robustness for small and occluded targets. The remainder of this paper is organized as follows. Section 2 describes the XRBS imaging system, the dataset acquisition process, and the annotation strategy. Section 3 presents the proposed YOLOv11-XRBS framework in detail, including its architectural refinement, SaFL, and the recomposed loss function. Section 4 reports the experimental results, ablation studies, and comparative analyses, followed by an in-depth discussion of the findings. Finally, Section 5 concludes the paper and outlines directions for future research.

## 2. Dataset Construction

### 2.1. XRBS Imaging System

The XRBS system employs a flying-spot scanning mechanism combined with beam modulation, as illustrated in Figure 1a. A cone-shaped X-ray beam is generated by the source and subsequently collimated using a slit and a rotating chopper wheel to form a narrow, modulated beam. Vertical scanning is achieved by sweeping this beam across the object, while horizontal scanning is accomplished through the relative motion between the object and the imaging system. Based on the principle of Compton scattering, the system is particularly sensitive to low-Z elements—such as carbon, hydrogen, oxygen, and nitrogen—which are prevalent in explosive materials, thereby enabling effective detection.

In this study, a high-energy X-ray source (150 kV, 150 W) is employed to generate the scanning beam. The X-ray tube, operating at 150 kV, produces a continuous spectrum with a maximum spectral intensity of approximately 70–80 keV. Backscattered photons detected in the system typically fall within the 90–120 keV range, consistent with the Compton scattering characteristics of low-Z materials. These photons are collected by a plastic scintillator array coupled with wavelength-shifting fibers, which convert the radiation into visible light and guide it to photodetectors for electronic signal conversion. The resulting signals are digitized and processed to form grayscale XRBS images that primarily reflect object contours and scattering intensity. This setup enhances the collection efficiency of weak backscattered signals and improves the quality of images. An experimental platform was constructed in the laboratory, as shown in Figure 1b; this incorporated the X-ray source, beam modulation components, detector modules, and shielding structures. Key system parameters are summarized in Table 1. A more detailed design of the scintillator-based detector system can be found in our previous work [20].

### 2.2. Dataset Acquisition

As publicly available X-ray datasets are predominantly based on transmission imaging, annotated datasets for XRBS remain limited. In this study, 2000 raw XRBS images were acquired using the proposed intelligent imaging system, featuring ten types of simulated explosives (Figure 2). These simulated explosives are not active explosives but inert models that are only used for research and training purposes. Each image contains between one and five explosive objects. The simulated explosives—constructed from metal or plastic and partially filled with polymethyl methacrylate granules and 5 mm steel balls—were categorized into four shape-based classes: general-shaped (GN), rectangular (GR), other-shaped (GO), and handle-shaped (GH).

To simulate realistic security inspection scenarios, various distractor objects (e.g., metal tanks, plastic containers, cups) and concealment devices (e.g., cardboard boxes, wooden crates, suitcases) were included, generating diverse occlusions and cluttered backgrounds (Figure 3). Occlusion and backgrounds were physically recreated during the imaging process by placing explosives alongside ordinary items or concealing them inside containers of different thicknesses. This design ensured that the dataset reflects realistic inspection environments with overlapping objects and visually complex scenes rather than artificially synthesized backgrounds.

All images were resized to 640 × 640 pixels and annotated using oriented bounding boxes, defined by the center coordinates, width, height, and rotation angle. A custom annotation tool was developed to ensure precise annotation. In total, the dataset contains over 10,000 annotated XRBS images derived from the raw acquisitions and augmented samples. Approximately 85% of these images contain one to five explosive-like objects—mainly inert replicas of GN, GR, and GH grenades—while the remaining 15% contain no explosives to support the evaluation of false alarms. The dataset was divided into training (70%), validation (10%), and testing (20%) subsets. All images were manually annotated with oriented bounding boxes using LabelImg (Version: 1.8.6). The acquisition process involved multiple concealment scenarios, including open exposure, partial occlusion, and packing within common luggage items, thereby increasing the realism of the dataset and its suitability for security inspection research.

To improve the generalization of the model, various data augmentation techniques—rotation, flipping, scaling, and noise injection—were applied [21]. To further simulate real-world concealment scenarios, common objects such as cups, bottles, and computer mice were used to partially obscure explosives during image acquisition. Additionally, random occlusion patches were added post-capture to increase the complexity of the dataset and its visual diversity. To prevent overfitting during training, an early stopping strategy was employed. Training was halted if the validation *mAP* failed to improve for 20 consecutive epochs, and the checkpoint with the highest validation *mAP* was preserved for subsequent evaluation. These measures ensured the reliable performance of the model and improved its generalization capabilities.

### 2.3. Image Processing

XRBS images often exhibit scattered pixel noise and block artifacts due to detector interference, uneven radiation exposure, and environmental conditions. These distortions obscure salient features and significantly reduce the accuracy of identification. To mitigate such noise, four filtering techniques were evaluated: mean filtering, median filtering, Gaussian filtering, and bilateral filtering. Representative results of each method are illustrated in Figure 4.

To quantitatively compare the filtering methods, three image quality metrics were used: the peak signal-to-noise ratio (*PSNR*) [22], the structural similarity index measure (*SSIM*) [23], and learned perceptual image patch similarity (*LPIPS*) [24]. The corresponding calculation formulas are provided in Equations (1)–(3):(1)$$PSNR=20\cdot \log_{10}\left(\frac{MAX_{1}}{\sqrt{MSE}}\right)$$(2)$$SSIM=\frac{\mu_{f}\mu_{g}+C_{1}}{\mu_{f}^{2}+\mu_{g}^{2}+C_{1}}\cdot \frac{\sigma_{f}\sigma_{g}+C_{2}}{\sigma_{f}^{2}+\sigma_{g}^{2}+C_{2}}\cdot \frac{\sigma_{fg}+C_{3}}{\sigma_{f}\sigma_{g}+C_{3}}$$(3)$$LPIPS=\sum_{l}\frac{1}{H_{l}W_{l}}{\sum_{h,w}\left\|w_{l}⊙\left({\overset{\wedge }{y}}_{hw}^{l}-\wedge {\overset{\wedge }{y}}_{0hw}^{l}\right)\right\|}_{2}^{2}$$

Here, *MSE* denotes the mean squared error between the grayscale values of the original and processed images; *μ_f_* and *μ_g_* represent the mean values of the original and processed images, respectively; *σ_fg_* is the covariance between the two; and *C_i_* is a stability constant. For mean, median, and Gaussian filtering, a 3 × 3 kernel was applied. Bilateral filtering was performed using radii of 30 and 50. To ensure general applicability, three representative XRBS images were selected, and the metrics were computed individually. The evaluation results are summarized in Table 2.

The results indicated that bilateral filtering consistently achieves the highest *PSNR* and *SSIM* scores and the lowest *LPIPS* values across all test samples. This demonstrates its superior ability to suppress noise while preserving both structural integrity and perceptual quality. Accordingly, bilateral filtering was selected as the denoising method for the SBCXray dataset to enhance the robustness of subsequent identification algorithms.

## 3. Algorithm Optimization

### 3.1. Adaptive Architecture Refinement

Although YOLOv11 demonstrates strong general-purpose identification capabilities, its baseline configuration is not tailored to the identification of small, low-detail explosive objects in complex XRBS backgrounds [25]. To address this limitation, we introduce an adaptive structural refinement strategy that systematically adjusts the convolutional hierarchy to improve the granularity of feature extraction. Specifically, the network depth was moderately increased to enrich semantic representation, while the convolutional kernel size was expanded from 3 × 3 to 5 × 5 to enlarge the receptive field and capture fine-scale spatial variations (Figure 5). This dual adjustment allows the network to balance global contextual perception with localized detail extraction, thereby enhancing its ability to discriminate small and ambiguous targets.

### 3.2. Size-Aware Focal Loss (SaFL)

The identification of small explosives is inherently hindered by scale imbalance: smaller objects are easily overshadowed by background clutter or dominant large-scale targets. To mitigate this, we reformulate the training objective by introducing Size-Aware Focal Loss (SaFL). Building upon the classical Focal Loss [26], SaFL incorporates a scale-sensitive weighting mechanism that dynamically emphasizes the contribution of small objects during training.

The loss is expressed as follows:(4)$$FL(p_{t})=-\alpha {\left(1-p_{t}\right)}^{\gamma }\log(p_{t})$$
where *p_t_* is the predicted probability for the true class, *α* is a balancing factor used to address class imbalance, and *γ* is the focusing parameter, typically set to *γ* = 2, to increase emphasis on difficult samples.

By applying Focal Loss, the model shifts attention toward small and difficult-to-detect explosive targets during training. The modulation term (1 − *p_t_*) *^γ^* suppresses the loss contribution from well-classified (often large) objects, allowing the network to better learn subtle features of small explosives. This modification transforms Focal Loss from a generic imbalance-handling strategy into a task-specific objective function tailored to XRBS explosive identification, thereby reflecting the methodological innovation of this work.

### 3.3. Loss Function Recomposition

The standard YOLOv11 employs a composite loss function that integrates classification loss (*L_cls_*), localization loss (*L_loc_*), and confidence loss (*L_conf_*). While effective in generic object identification tasks, this formulation inadequately captures the precision requirements of small-object localization. To address this, we propose a recomposed loss framework that introduces scale-adaptive penalties to magnify the learning signal associated with small objects. The total objective function is defined as follows:(5)$$L_{total}=L_{cls}+L_{loc}+L_{conf}$$
where *L_cls_* quantifies the discrepancy between predicted and true class labels, *L_loc_* measures the error in bounding box regression (typically using *MSE*), and *L_conf_* evaluates the alignment between the predicted confidence scores and object presence.

To further enhance small object identification, weighted factors were introduced into the localization loss, increasing the penalty for inaccuracies in small object predictions. This encourages the model to allocate more attention to small objects during training. The modified localization loss is defined as follows:(6)$$L_{total}=L_{cls}+\alpha \cdot L_{loc}+\beta \cdot L_{conf}$$
where *α* and *β* are weighting factors adjusted according to object sizes. For small objects, higher values of *α* and *β* increase their contribution to the total loss. Additionally, an IoU-weighted term was incorporated to refine localization loss [27]:(7)$$L_{loc}=\sum_{i=1}^{N}\left(1-IoU\left(b_{i},{\hat{b}}_{i}\right)\right)\cdot \omega_{i}$$
where *b_i_* and ${\hat{b}}_{i}$ represent the ground truth and predicted bounding boxes for the i-th object, respectively, and *w_i_* is a size-dependent weight. These enhancements prioritize spatial precision for small objects. The effectiveness of these loss function modifications is evaluated through ablation experiments in later sections. Through this recomposed formulation, the loss function is transformed into a scale-sensitive optimization framework, directly aligning the training dynamics with the challenges of small explosive identification in XRBS imagery.

## 4. Results and Discussion

### 4.1. Ablation Study

To evaluate the effectiveness of each proposed optimization, an ablation study was performed using the SBCXray dataset. Beginning with the baseline YOLOv11 model, enhancements were introduced incrementally—namely, Focal Loss, weighted localization loss, and architectural modifications—to assess their individual and combined contributions to detection performance. The results are presented in Table 3. A closer examination of Table 3 reveals how each component contributes to performance enhancement. The architecture refinement alone improves recall and *mAP* by enlarging the receptive field and capturing multi-scale contextual information, which is essential for recognizing low-detail explosive contours. The addition of SaFL further boosts recall by emphasizing small and difficult samples during training, directly reducing missed detections. Finally, the recomposed loss framework improves precision and the accuracy of localization by penalizing misaligned bounding boxes more effectively. The cumulative effect of these improvements is a significant increase in the F1 score and *mAP*, demonstrating that the proposed modifications are not generic but specifically designed to meet the unique challenges of XRBS imagery.

To assess the overall effectiveness of the proposed enhancements, all three components—loss function adjustments, network structure modifications, and small-object detection strategies—were integrated into a unified model, referred to as YOLOv11-XRBS. Comparative experiments on the SBCXray dataset demonstrated consistent performance improvements across multiple evaluation metrics. Notably, the model showed enhanced detection for small and visually challenging explosive objects, particularly those with irregular shapes or partial occlusion. These results indicate that the combined optimizations significantly improve the model’s robustness and accuracy in complex detection scenarios. However, as the components were applied jointly, their individual contributions remain intertwined. Future work will include a more granular ablation analysis to isolate and quantify the marginal effects of each strategy, enabling the more targeted refinement of the model in subsequent iterations.

### 4.2. Performance Comparison

The optimized YOLOv11-XRBS model outperforms previous YOLO variants, including YOLOv9t [28] and YOLOv10n [29], across multiple performance metrics. Notably, it achieves higher scores in *Precision* (accuracy of positive predictions) [30], *Recall* (completeness of detection) [30], *F1* Score (harmonic mean of precision and recall) [30], and mean average precision (*mAP*) [31]. Their definitions are as follows:(8)$$Precision=\frac{TP}{TP+FP}$$(9)$$Recall=\frac{TP}{TP+FN}$$(10)$$F1=\frac{2\cdot Precision\cdot Recall}{Precision+Recall}$$(11)$$mAP=\frac{1}{N}\sum_{i=1}^{N}AP_{i}$$
where *TP*, *FP*, and *FN* denote true positives, false positives, and false negatives, respectively, and *AP_i_* is the average precision for the i-th class.

To further strengthen the comparison, we also evaluated two widely used non-YOLO frameworks, VGGNet [32], RetinaNet [26], DETR [33], Faster R-CNN [34] and SSD512 [35]. These results are summarized in Table 4.

As shown in Table 4, YOLOv11-XRBS achieves the highest *mAP* of 94.8%, outperforming all other methods. Faster R-CNN achieves *mAP* of 92.1%, but its performance is limited by its two-stage pipeline, which affects its ability to capture weak structural cues in low-detail XRBS images. SSD512, with *mAP* of 89.7%, struggles particularly with small targets, leading to a decrease in recall and overall *mAP*. RetinaNet, with *mAP* of 91.4%, performs better than SSD512, thanks to its Focal Loss, which helps improve small object detection, yet it still falls short of YOLOv11-XRBS in both precision and recall. DETR, achieving *mAP* of 93.6%, performs well in complex scenes but has trouble detecting small objects in low-resolution XRBS images. VGGNet, while effective in standard image classification, performs the worst with *mAP* of 86.3%, particularly in the detection of small or occluded objects. Overall, YOLOv11-XRBS stands out due to its balanced performance across all metrics, offering superior robustness in detecting small, low-contrast explosive targets typical in XRBS images.

To further analyze the performance of the model across different categories of explosive-like objects, Table 5 reports class-wise detection results. As shown in Table 5, YOLOv11-XRBS achieves the highest accuracy across all four shape categories (GN, GR, GO, and GH). The most significant performance gap is observed in the GH category, which represents handle-shaped explosives with irregular contours and fewer training samples. Faster R-CNN shows moderate performance but struggles with fine structural details in the GH category, while SSD512 exhibits a significant drop in accuracy (80.3%) for detecting small or irregular objects. RetinaNet performs better than SSD512, particularly in small object detection, but still falls short of YOLOv11-XRBS in precision and recall. DETR performs well in object localization but is less effective in detecting small, low-contrast objects like those in the GH category. VGGNet achieves the lowest accuracy across all categories, particularly in the detection of irregular objects. YOLOv11-XRBS benefits from its adaptive architecture and Size-Aware Focal Loss, improving robustness under occlusion and low-detail conditions. These results clearly demonstrate YOLOv11-XRBS’s superiority not only over YOLO variants but also compared to non-YOLO frameworks, underscoring its value for XRBS explosive detection.

Figure 6 provides a visual comparison of the detection results across various YOLO models, offering an intuitive understanding of their performance on the SBCXray dataset. The most prominent performance gap appears in the detection of GH explosives. For instance, YOLOv9t (fourth image) shows repeated detections for the bottom GH sample; in YOLOv11 (second image), the topmost GH sample is missed. Repeated detections are observed in both the first and fourth images for the same bottom GH object. For other explosive categories, YOLOv6n tends to misclassify distractor objects as GN or GR explosives under high visual confusion. Across all models, false positives and missed detections frequently occur when explosive objects overlap with background clutter, especially near image edges. To further quantify this effect, we conducted an additional analysis on a randomly selected set of 100 test images. The optimized YOLOv11-XRBS produced two false positives and one false negative, whereas the baseline YOLOv11 produced four false positives and four false negatives, corresponding to false positive rates of 2% and 4%, respectively. These results indicate that although false alarms do not compromise security, excessive false positives may reduce the efficiency of the operator and negatively affect the experience of passengers. In contrast, YOLOv11-XRBS consistently identifies explosive objects with higher confidence, even under challenging conditions, and performs accurate detection in images with low background complexity. These results highlight the model’s robustness to occlusion, shape variation, and noise, underscoring its suitability for real-world XRBS-based explosive detection tasks.

Beyond these observations, it is important to analyze why YOLOv11-XRBS achieves more reliable detection compared to the baseline models. The refinement of the adaptive architecture enhances multi-scale feature extraction and increases the effective receptive field, which improves the rate of detection of small and irregularly shaped explosives, especially under occlusion. The integration of SaFL explicitly increases the learning weight of small and difficult targets, which reduces the frequency of missed detections in cluttered or edge regions. Moreover, the recomposed loss function introduces scale-adaptive penalties, leading to more accurate bounding box localization. Taken together, these tailored improvements explain why YOLOv11-XRBS achieves fewer false positives and negatives, as shown in Figure 6, particularly in challenging XRBS scenarios where traditional YOLO variants struggle.

It should be noted that the present work focuses on explosive-like objects with relatively fixed and recognizable shapes, such as grenades. While this provides a solid foundation for validating the methodology, the current system is not yet capable of detecting improvised explosive devices disguised as ordinary household items or lacking distinctive contours. Future research will address this limitation by expanding the dataset to cover a wider variety of disguised explosive scenarios and by incorporating physical-layer detection techniques to analyze material composition. Such multimodal approaches are expected to enhance the robustness and applicability of XRBS-based explosive detection. It should also be emphasized that the proposed method does not completely eliminate the intrinsic limitations of XRBS imaging, such as its low contrast and weak structural detail, but it reduces their negative impact on recognition performance. Future work concerning image recognition for XRBS explosive detection should therefore aim to further overcome these intrinsic limitations by developing more advanced feature extraction strategies, integrating multimodal information sources, and leveraging emerging deep learning architectures to achieve more reliable and comprehensive detection performance.

## 5. Conclusions

This study investigated the application of deep learning methods to the recognition of explosive objects in XRBS images. A custom dataset, namely SBCXray, containing more than 10,000 annotated images of simulated explosives under different concealment conditions, was constructed to support the research. On this basis, an improved model, YOLOv11-XRBS, was developed with three enhancements: adaptive architectural refinement for better multi-scale feature representation, SaFL to improve the recognition of small and low-detail targets, and a recomposed loss function to enhance bounding box localization. These modifications were designed in response to the specific challenges of XRBS imaging and were evaluated in a series of comparative experiments.

The experimental results showed that YOLOv11-XRBS provided a more balanced performance in terms of precision, recall, F1-score, and *mAP* than the baseline YOLO variants and other non-YOLO frameworks, including Faster R-CNN, SSD512, RetinaNet, DETR, and VGGNet. At the same time, the current work remains limited to explosive-like objects with relatively fixed shapes, and the recognition of improvised explosive devices disguised as household items has not yet been addressed. Future work will extend the dataset and explore multimodal approaches that integrate material-level analysis in order to improve the generalizability of XRBS-based explosive recognition systems.

## Figures and Tables

**Figure 1 sensors-25-06130-f001:**
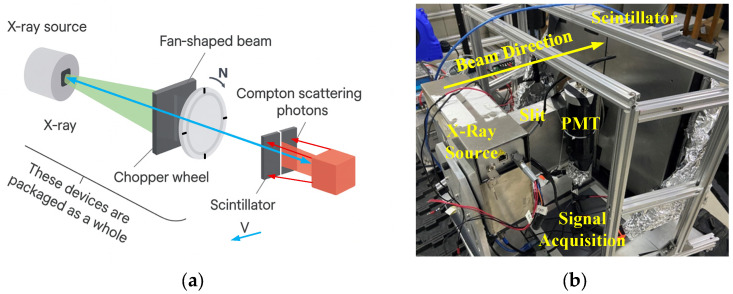
(**a**) Schematic of the XRBS Compton scattering detection system based on flying-spot scanning; (**b**) assembled laboratory prototype [20].

**Figure 2 sensors-25-06130-f002:**
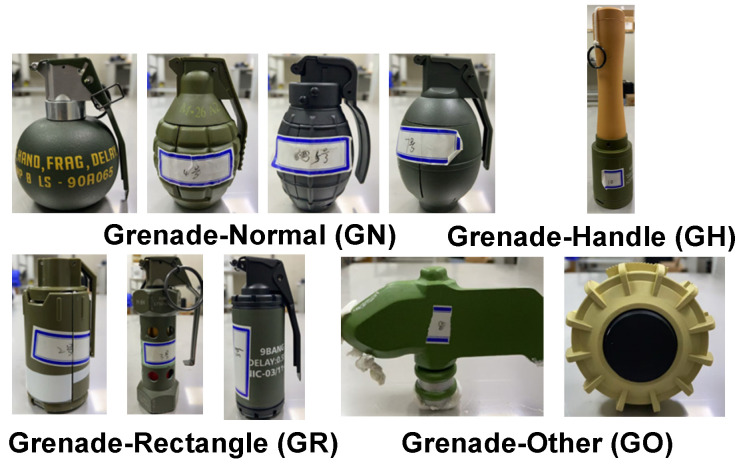
Four shape-based categories of simulated explosives used for dataset annotation.

**Figure 3 sensors-25-06130-f003:**
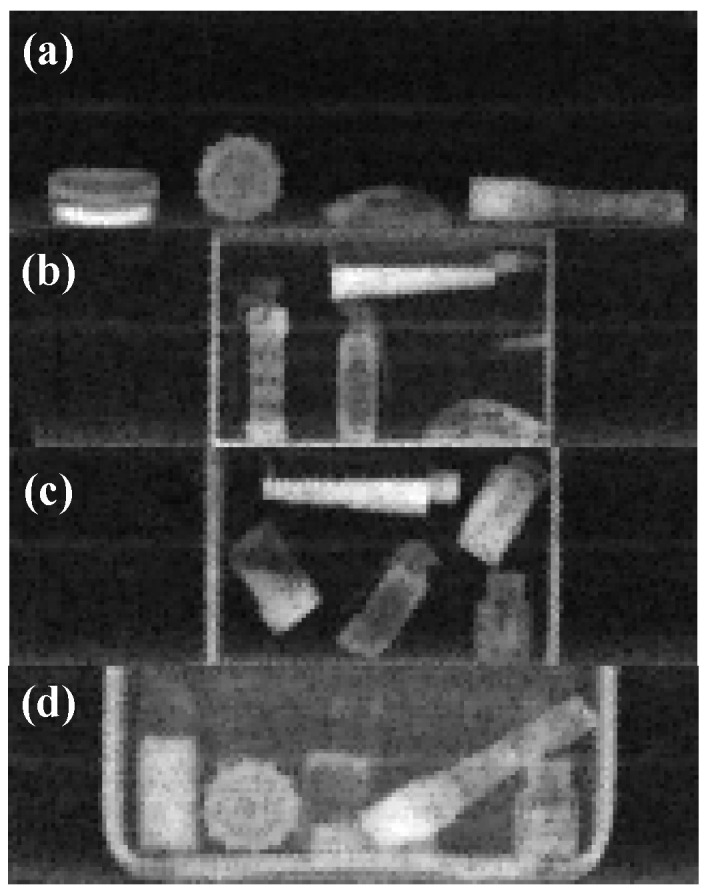
Example images captured under varying concealments: (**a**) no obstruction, (**b**) concealed in a wooden box (8 mm wall), (**c**) cardboard box (3 mm wall), (**d**) suitcase (2 mm wall).

**Figure 4 sensors-25-06130-f004:**
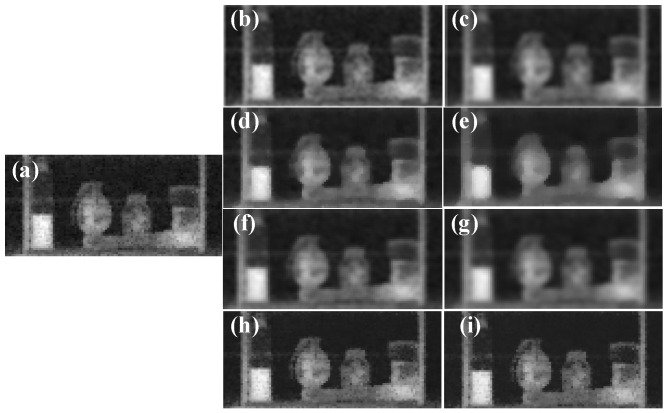
Images processed with four different filtering methods are shown. (**a**) Original image; (**b**,**c**) mean filtering (3 × 3, 5 × 5); (**d**,**e**) median filtering (3 × 3, 5 × 5); (**f**,**g**) Gaussian filtering (3 × 3, 5 × 5); (**h**,**i**) bilateral filtering (radii 30, 50).

**Figure 5 sensors-25-06130-f005:**
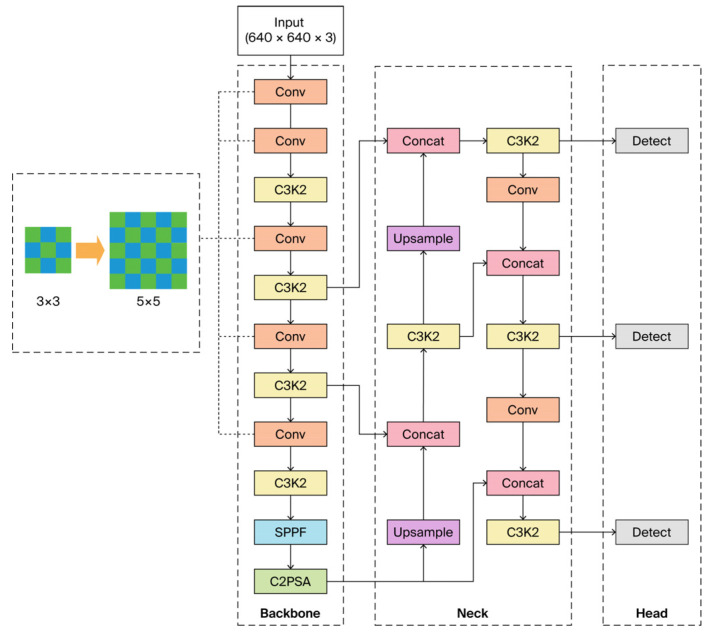
Structural refinements introduced into the YOLOv11 backbone to enhance multi-scale feature learning.

**Figure 6 sensors-25-06130-f006:**
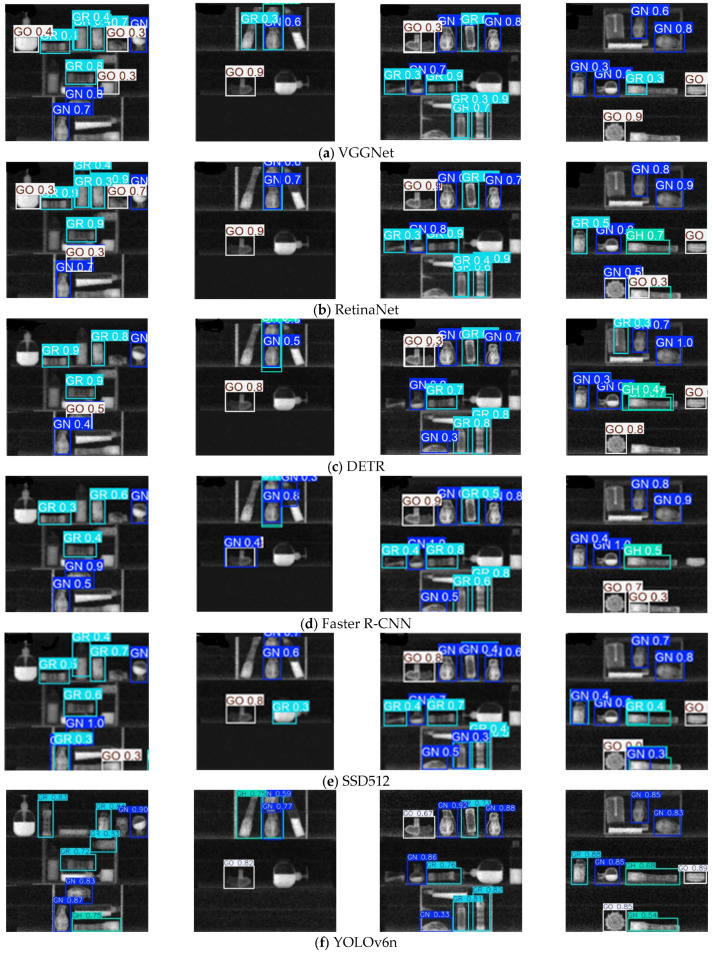
Visualization comparison of detection results for different YOLO models.

**Table 1 sensors-25-06130-t001:** Key parameters of the XRBS imaging system.

Component	Parameters
X-ray Source	150 kV, 150 W
Number of Detectors	2
Detector Size	400 mm × 200 mm × 50 mm
Spatial Resolution	1 mm
Field of View	>30 cm

**Table 2 sensors-25-06130-t002:** Quantitative evaluation of different filtering methods on XRBS images using *PSNR*, *SSIM*, and *LPIPS* metrics.

Filtering Methods	Figure	PSNR	SSIM	LPIPS
Mean Filtering	a	27.95	0.86	0.20
	b	28.08	0.88	0.22
	c	27.60	0.88	0.24
Median Filtering	a	29.45	0.88	0.09
	d	30.21	0.90	0.06
	e	29.31	0.91	0.10
Gaussian Filtering	a	29.90	0.91	0.14
	f	30.17	0.92	0.15
	g	29.60	0.92	0.19
Bilateral Filtering	a	33.09	0.94	0.03
	h	34.13	0.94	0.02
	i	34.54	0.98	0.01

**Table 3 sensors-25-06130-t003:** Ablation study of individual improvements on YOLOv11.

Variant	Precision (%)	Recall (%)	mAP (%)
Baseline YOLOv11	91.0	92.1	93.8
Add SaFL	91.3	92.3	94.1
Add SaFL and Loss Function Recomposition	91.9	92.8	94.5
Add SaFL, Loss Function Recomposition and Adaptive Architecture Refinement	92.9	93.5	94.8

**Table 4 sensors-25-06130-t004:** Comprehensive performance of different YOLO models on the SBCXray dataset.

Model	Precision	Recall	F1 Score	mAP (%)	COCO mAP (%)
VGGNet	84.3	84.9	85.7	86.3	63.8
RetinaNet	89.1	89.7	90.1	91.4	66.2
DETR	90.7	91.7	92.5	93.6	69.3
Faster R-CNN	90.3	91.0	90.8	92.1	66.2
SSD512	87.2	87.8	88.1	89.7	65.7
YOLOv6n	91.6	92.1	91.8	93.0	68.0
YOLOv8n	92.7	93.1	92.9	94.3	68.6
YOLOv9t	92.3	92.4	92.3	93.4	66.6
YOLOv10n	86.8	85.9	86.3	91.0	66.6
YOLOv11	91.0	92.1	92.7	93.8	68.4
YOLOv11-XRBS	92.9	93.5	93.2	94.8	72.2

**Table 5 sensors-25-06130-t005:** Per-category comparison on SBCXray dataset.

Model	mAP (%)	AP			
		GN	GR	GO	GH
VGGNet	86.3	91.2	88.4	86.7	79.5
RetinaNet	91.4	96.8	94.1	91.3	83.7
DETR	93.6	97.4	98.4	93.5	85.1
Faster R-CNN	92.1	97.3	95.8	90.3	85.6
SSD512	89.7	95.3	92.2	90.3	80.3
YOLOv6n	93.0	94.8	91.2	92.5	87.9
YOLOv8n	94.3	97.6	97.0	94.4	88.3
YOLOv9t	93.4	97.0	96.3	92.7	87.5
YOLOv10n	91.0	97.0	94.9	91.0	81.0
YOLOv11	93.8	97.1	97.7	92.8	88.6
YOLOv11-XRBS	94.8	98.1	98.6	93.3	89.6

## Data Availability

Data available on request due to privacy or ethical restrictions. The data presented in this study are available on request from the corresponding author due to the restriction that the dataset, which is primarily used for explosive object detection, may be misused for non-civilian purposes.

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
