# Peer review of "YOLOv11-XRBS: Enhanced Identification of Small and Low-Detail Explosives in X-Ray Backscatter Images"

_sensors, 2025, doi:10.3390/s25196130_

Round 1
Reviewer 1 Report
Comments and Suggestions for Authors
This paper primarily proposes an improved model based on the YOLOv11 architecture (YOLOv11-XRBS) for detecting explosives in X-ray backscatter (XRBS) images. However, the paper lacks originality, as it mainly combines existing methods without providing novel contributions. Additionally, the overall writing is not sufficiently rigorous. The specific issues are as follows:
-
Lack of Innovation: The method presented in the paper mainly combines existing research results. For example, in Section 3.1, the authors only modify the size parameters of certain convolutional layers in YOLOv11, without introducing any new network modules. Furthermore, there is no analysis of the rationale behind the parameter modifications, which makes this part of the work superficial. In Section 3.2, the introduction of Focal Loss is based on existing research, and the authors do not demonstrate any originality in this part. While Focal Loss helps with small object detection, it is not an original contribution, and thus does not reflect the independent contribution of this paper.
-
Insufficient Experimental Design and Comparison: Although the paper mentions comparisons with other YOLO variants, it does not provide a comprehensive comparison with other existing explosive detection methods. Comparing only with certain YOLO variants does not fully demonstrate the superiority of the proposed method. There is a lack of sufficient experimental validation to support the effectiveness of the model.
-
Insufficient Results and Discussion: The results and discussion section merely describes the experimental outcomes without providing a deep analysis of how the model's performance was improved. The paper does not address how the decision-making process of the model is explained, nor does it clarify how the proposed improvements contribute to performance enhancement.
-
Low-Level Writing Errors: There are several low-level errors in the paper, such as the title of Table 2 still being the original template title, which is unrelated to the content of the paper, and the repetition of data in the 2nd and 6th rows of Table 3. Such errors significantly impact the overall quality of the paper.
If the authors fail to clearly identify the innovations of this paper, provide rigorous and logically sound experimental analysis, and correct the low-level writing errors in the revised version, then this paper lacks sufficient academic contribution and does not meet the standards for publication in an SCI journal.
Reviewer 2 Report
Comments and Suggestions for Authors
The presented in the paper results demonstrate that the proposed optimizations significantly improve the detection of small, low-detail, and occluded explosive objects. The demonstrated YOLOv11-XRBS model integrates focal Loss, oriented bounding box regression, and architectural enhancements within the YOLOv11 framework. As a result, YOLOv11-XRBS meets the practical performance requirements for high-throughput, field-deployable XRBS systems. The results obtained in the paper are important and useful for applications in the field of intelligent explosive detection using XRBS-based security inspection systems. The manuscript is well-written and clear understandable. I could recommend its possible publication in Sensors, however, there is a couple of question that should be addressed prior to a publication.
1. What was the energy range (wavelengths) of the incident and scattered photons, at least approximately? This characterizes the radiation source in more detail, as well as the process of Compton scattering on the objects being studied.
2. When one consider training, which loss function (citation of formula is required) has been used during training: train/val/test sizes, stopping criterion to prevent overfitting?
Reviewer 3 Report
Comments and Suggestions for Authors
The authors conducted a study that is undoubtedly of great practical importance. However, for publication they need to rework the text of the article.
- The authors need to formulate the goals of scientific research more precisely. In the introduction, they write about improvised explosive devices and the search for explosives. The main part presents quite factory-made explosive devices (hand grenades of various types).
- Then it turns out that the search is specifically for objects with such a specific shape. However, improvised explosive devices are usually disguised as household items and the presented algorithm will not detect them. This does not detract from the importance of the article, since hand grenades can be put in luggage simply by not very smart passengers and this also needs to be detected, but the text of the article needs to be corrected.
- In my opinion, the main metric for this kind of system is reliability. This is very important! The number of frames processed per second should be equal to or slightly higher than the frequency of a standard video stream, and given the operating mode of such scanners, this frequency may be even lower! The authors are completely wrong to stick out this metric.
- It is not very clear from the text of the article whether each image contains a dummy grenade. If each one, then another question arises, how high will the frequency of false alarms of the system be when the presence of a grenade is detected in the luggage, where it is not. On the one hand, an error of this kind does not affect safety, on the other hand, the increased frequency of false alarms is not well perceived by both security services and passengers. Therefore, I would like to get an estimate of this parameter in the article.
Reviewer 4 Report
Comments and Suggestions for Authors
Title: YOLOv11-XRBS: Enhanced Detection of Small and Low-detail 2 Explosives in X-ray Backscatter Images
In this work proposed a You Only Look Once version 11 (YOLOv11)-XRBS, an enhanced version of YOLOv11 tailored for explosive detection in XRBS images. A high-quality dataset (SBCXray) comprising over 10,000 annotated images of diverse simulated explosive scenarios was developed to support training and evaluation. The work is good and interesting in its field; however, this manuscript currently contains major and minor corrections (shown below), which should be carefully considered.
Remarks to the Authors: Please see the full comments.
1-A section of the related works is required to discuss different recent detection methods of Explosives in X-ray Backscatter Images based on their pros and cons.
2-What are the specific problems that are solved in the current work? Several problems were mentioned in the Abstract (low image contrast, background noise, small object size, and lack of visual detail). Have all these problems been solved in the proposed work? Please explain clearly.
3-It is stated that “A custom dataset comprising over 10,000 annotated XRBS images of simulated explosive scenarios under varied concealment conditions was constructed.” More details about this dataset are required.
4-For more organization, the structure of the paper can be added at the end of the introduction section.
5-The Research Contributions should be listed and highlighted clearly since there are many good points in the proposed research.
6-Regarding subsection 2.1 and Figure 1, there is no clear mention of the original references used in the figure [Research on Wavelength-Shifting Fiber Scintillator for Detecting Low-Intensity X-Ray Backscattered Photons]. Furthermore, this section should describe the main steps of the proposed methodology without relying on the work of previous authors (even if they are the same authors).
7-How are the various occlusions and busy backgrounds generated? Please explain.
8-Any information, graph, equation, or data set taken from a previous source must be documented with a reliable source, unless it belongs to the authors. Please check this issue for the full manuscript.
9-In general, the organization of the research should be improved (the arrangement of sections and the workflow).
10- Data splitting is mentioned where, it was divided into training (70%), validation (10%), and testing (20%) subsets. How the validation process was done. Please give more details.
11- How are the weighting factors adjusted according to object sizes for the loss function?
12- The mathematical formula of evaluation metrics needs to be added to the manuscript even they are well known.
13- Table 3 should be more organized to be clearer to the reader. Additionally, why do the first and last rows of the table contain the same results?
For all Tables, please highlight the best results in Bold.
Comments on the Quality of English LanguageThe English could be improved to more clearly express the research.
Round 2
Reviewer 1 Report
Comments and Suggestions for Authors
After carefully reviewing the revised version, I believe that the paper has improved in terms of originality, experimental design, and result analysis, and its overall quality has been enhanced. However, there are still a few minor issues that need further refinement. Therefore, I recommend accepting the paper, but with some minor revisions. Below are my specific suggestions:
-
The background introduction to XRBS images has been improved, but it could still be further expanded. When discussing existing methods, it would be helpful to provide a more detailed comparison with other fields (such as X-ray transmission images and infrared images), especially regarding the challenges related to small objects and low-resolution images.
-
The experimental design has improved, but I still suggest strengthening the comparison with other traditional explosive detection methods, particularly non-YOLO frameworks. Although comparisons with Faster R-CNN and SSD512 have been added, it would be beneficial to include some classic methods from other fields to further validate the advantages of YOLOv11-XRBS across different tasks.
Reviewer 3 Report
Comments and Suggestions for Authors
The authors corrected the article based on my comments. Overall, the article began to look more logical. I think it can be published.
Author Response
I would like to sincerely thank you for your insightful and valuable feedback during the first round of review. Your suggestions helped significantly enhance the quality and clarity of the manuscript, and I believe your input has made the paper stronger and more focused.
I also deeply appreciate your positive evaluation of the revised manuscript. Your recognition of the improvements made is highly encouraging, and I am grateful for your support in recommending the paper for acceptance.
Reviewer 4 Report
Comments and Suggestions for Authors
Title: YOLOv11-XRBS: Enhanced Identification of Small and Low- detail Explosives in X-ray Backscatter Images
Most of the comments have been addressed; however, there are some minor points should be considered which are as follows:
1-For Figure1, there is no need to write the title of the reference in the caption.
2-The format of writing the references should be checked carefully.
3- It is recommended that you review the grammar in the writing again, please.
Comments on the Quality of English LanguageThe English could be improved to more clearly express the research.
